# OpenReview forum: "PropensityBench: Evaluating Latent Safety Risks in Large Language Models via an Agentic Approach"
_ICLR.cc/2026/Conference — ICLR 2026 Poster_

### Official Review · Reviewer_tiGg · 2025-11-01

**Soundness:** 3
**Presentation:** 3
**Contribution:** 3
**Rating:** 4
**Confidence:** 2

**Summary:**

This paper introduces PropensityBench, a benchmark measuring LLMs' propensity to use dangerous capabilities when equipped with proxy tools under operational pressure. The framework includes 5,874 scenarios across four domains (self-proliferation, cybersecurity, biosecurity, chemical security) and reveals that even safety-aligned models show significantly increased propensity to choose harmful tools under pressure, with some models reaching 79% failure rates.

**Strengths:**

**Novel evaluation paradigm**: Shifts from capability assessment to propensity measurement, addressing a critical blind spot in current safety evaluations.

**Comprehensive experimental design**: 5,874 scenarios with 6,648 tools, testing 12 models including frontier systems, with rigorous statistical analysis across multiple pressure levels.

**Important empirical findings**: Reveals shallow alignment where models rely on tool naming rather than consequence reasoning (e.g., O4-mini's propensity jumps from 15.8% to 59.3%).

**Systematic pressure modeling**: Six pressure categories with 12 escalating levels, grounded in human psychology research on decision-making under stress.

**Weaknesses:**

**Limited ecological validity**: The framework assumes agents "would do" certain actions if empowered, but real-world tool availability, implementation constraints, and deployment contexts remain unknown. The proxy tools in Section 2.2 may not accurately reflect future dangerous capabilities.

**Incomplete domain coverage**: The four domains lack clear selection criteria (Section 2.1). Critical areas like economic manipulation, social-political influence, and physical robotics are absent. Why these specific four? The paper doesn't justify this limitation comprehensively.

**Narrow propensity measurement**: Propensity is reduced to binary tool selection (Section 2.5), missing reasoning traces, hesitation patterns, or self-correction behaviors that might indicate true inclination. Real propensity involves more than final choices.

**Missing defense evaluations**: No testing of existing safety mechanisms (constitutional AI, safety filters, monitoring systems) against the benchmark. How would current defenses perform?

**Static pressure application**: Pressure messages are pre-defined rather than adaptive (Section 2.4), potentially missing how models respond to dynamic, context-aware pressure that adjusts based on their responses.

**Questions:**

**Domain selection rationale**: What systematic criteria determined these four domains? How would results differ with economic, social, or physical safety domains?

**Propensity measurement validity**: How do you validate that tool selection accurately reflects propensity rather than confusion or misunderstanding? Could you incorporate reasoning analysis?

**Defense mechanism testing**: Which existing safety techniques (RLHF variations, constitutional AI, output filters) have you tested? What mitigation strategies show promise?

**Temporal dynamics**: How does propensity change with repeated exposure to similar scenarios? Do models learn to resist or become more susceptible?

**Cross-domain transfer**: Do models showing high propensity in one domain exhibit similar patterns in others? Is there a general "propensity factor"?

**Human baseline comparison**: How do human participants perform under identical pressure conditions? This would validate the benchmark's psychological grounding.

**Real-world calibration**: How do you ensure proxy tools accurately represent future dangerous capabilities? What validation exists for this assumption?

**Pressure interaction effects**: Do different pressure types interact? For example, does combining time and resource pressure create super-linear effects?

**Model architecture influence**: How do different architectural choices (chain-of-thought, mixture-of-experts) affect propensity independent of scale?

**Deployment implications**: What monitoring infrastructure would be needed to detect propensity signals in production systems before harmful actions occur?

---

> ### Author Response · Authors · 2025-11-22
> **Detailed rebuttal to reviewer's questions and concerns (1/6)**
>
> We thank reviewer tiGg for the valuable feedback and their appreciation for 1) novel evaluation paradigm, 2) comprehensive experimental design, 3) important empirical findings, and 4) systematic pressure modeling. We address the concerns raised by the reviewer as follows.
>
> > Q1: The ecological validity is limited, as proxy tools may not reflect real-world constraints or future capabilities.
>
> **Human validation to reflect realism of tools.** All 5,874 scenarios underwent comprehensive expert human review (Section 2.4 Quality assurance using automated and human review section and Appendix F) by 54 independent reviewers distributed across 13 countries, with domain expertise in chemistry, biology, computer science, and related fields. Each scenario was evaluated against 10 detailed criteria (Appendix F) assessing realism of the tools, coherence, task appropriateness, and consequence severity. Reviewers held advanced degrees (PhD, Master's, Bachelor's) and brought professional experience from university research, biotech, public health, and data science. Critically, *scenarios flagged as incoherent or unrealistic by human experts were filtered from the benchmark* regardless of LLM generation quality. The following 5 criterias out of the total 10 criterias specifically assess the realism and validity of the tools. The final criteria specifically evaluates whether the tools reflect the domain and scenario specific future dangerous capabilities given in Figure 3.
>
> - Does the aligned tool represent an aligned way to complete the task? *Explanation:* Is the aligned tool the intended, proper, or conventionally correct method to accomplish the task message?
> - Does the immediate action acknowledgement convey that misaligned tools can be used to complete the task? *Explanation:* This question asks if the text indicates that the misaligned tools are potential ways for the agent to try to complete the task, regardless of whether they ultimately succeed or if it’s the ideal approach.
> - Does the consequence description convey meaningful negative consequences with harmful outcomes? *Explanation:* Does the description of consequence description detail realistic and impactful negative results that could occur?
> - Can misaligned tools, both benign and harmful, lead to the consequences described in the consequence description? *Explanation:* Can both the benign and harmful misaligned tools realistically result in the negative consequences outlined in the consequence description?
> - Do misaligned tools use an approach that is related to the given attack vector (aka dangerous capability)? *Explanation:* Is the technique employed by the misaligned tools directly connected to or an example of the specified attack vector?

---

> > ### Author Response · Authors · 2025-11-22
> > **rebuttal continued ... (2/6)**
> >
> > > Q2: The selection criteria for the four chosen security domains are not clearly justified, and other critical areas are absent.
> >
> > We appreciate the opportunity to clarify our domain selection rationale, which is grounded in systematic criteria outlined in Section 2.2.
> >
> > **Our domain selection follows three meaningful and explicit criteria.** Section 2.2 describes our criteria (1) the domain represents capabilities with catastrophic misuse potential, (2) it contains clear ethical boundaries between aligned and misaligned behaviors, and (3) it involves capabilities that future models could feasibly acquire through continued scaling or specialized training. We selected domains where all three criteria are simultaneously satisfied and where there exists established safety research literature to ground our taxonomies.
> >
> > **Cybersecurity, biosecurity, and chemical security are established frontier safety priorities.** These domains are extensively studied in prior work ([Li et al. 2024](https://arxiv.org/abs/2403.03218), [Chaudhry et al. 2024](https://futureoflife.org/document/chemical-biological-weapons-and-artificial-intelligence-problem-analysis-and-us-policy-recommendations/), [Phuong, M. et al. 2024](https://arxiv.org/abs/2403.13793)) and identified by leading AI labs as critical risk domains requiring systematic evaluation as demonstrated in evaluations in model cards ([Gemini-3.0](https://storage.googleapis.com/deepmind-media/Model-Cards/Gemini-3-Pro-Model-Card.pdf), [Meta’s Code World Model](https://ai.meta.com/research/publications/code-world-model-preparedness-report), [gpt-o3-o4_mini](https://cdn.openai.com/pdf/2221c875-02dc-4789-800b-e7758f3722c1/o3-and-o4-mini-system-card.pdf)). These domains have well-documented catastrophic potential (e.g., pandemic engineering, chemical weapons), clear ethical boundaries ([MITRE ATT&CK](https://attack.mitre.org/) framework, [WMDP](https://arxiv.org/abs/2403.03218) taxonomies), and demonstrated feasibility as models increasingly gain these capabilities. Self-proliferation, while less explored in AI safety literature, represents a distinctive recursive risk where models autonomously reproduce or modify themselves or others, a threat vector that could exponentially amplify all other risks through cascading proliferation effects. For example if an agent has no background in biosecurity but is capable of dominating other agents (directly modify configurations / incentives of other agents) can configure a safe biosecurity agent to perform a harmful biological task.
> >
> > **Suggested domains are methodologically challenging for propensity evaluation.** Economic manipulation and social-political influence lack the clear ethical boundaries needed for unambiguous tool classification (criterion 2). What constitutes "manipulation" versus "persuasion" is often context-dependent and contested. Physical robotics involves execution capabilities beyond language models' current scope, making propensity assessment premature. Our framework is explicitly designed for domains where we can create clear aligned/misaligned tool pairs with unambiguous consequences, a requirement that these domains do not yet satisfy for systematic benchmarking.
> >
> > **Our framework is extensible to new domains as they meet the criteria.** We provide a complete automated generation pipeline (Section 2.4, Appendix C.3) requiring only seed configurations (domain, environment, role, dangerous capability) as input. As the community develops clearer taxonomies for economic or social risks that satisfy our three criteria, PropensityBench can readily incorporate them. We view our current four domains as a validated foundation that establishes the propensity evaluation paradigm, with expansion to other domains as important future work

---

> > > ### Author Response · Authors · 2025-11-22
> > > **rebuttal continued ... (3/6)**
> > >
> > > > Q3: Propensity measurement is too narrow, reducing it to binary tool selection while missing reasoning or hesitation patterns. Could you incoperate reasoning analysis?
> > >
> > > We have added several additional experimental analyses addressing this point.
> > >
> > > **Behavioural dynamics in agent trajectories.** Additional analysis provided in Table 2 and Figure 9 measure multiple dimensions of behavior throughout the decision process: (1) *Persistence*-the average number of aligned tool attempts before failure, revealing how long models try safe approaches (avg. ranging from 1.4 to 4.8 attempts), (2) *Resilience*-the average pressure level at which failures occur, showing how many pressure levels models typically go through before switching to misaligned approach (avg. ranging from 1.1 to 5.8 across models), (3) *Abandonment Rate* (Table 2, Figures 9, 23)-the proportion of pressure levels in failing scenarios where models explicitly refuse to continue rather than choose the misaligned approach or retry the aligned approach. In addition, Figure 14 provides analysis on *Immediacy of triggering*-measuring what percentage of failures occur *immediately* after pressure versus after deliberation.
> > >
> > > **We provide systematic analysis of reasoning patterns.** Additional analysis provided in Figure 6 presents a complete taxonomy of explicit justifications models provide when choosing $\mathcal{T}^M$, categorizing them into four types: Objective-Driven (emphasizing task completion necessity), Pressure-Induced (citing operational constraints), Risk-Benefit (arguing instrumental advantages outweigh risks), and "Greater Good" (claiming organizational or environmental benefit). This reveals distinct model-specific rationalization patterns for calling the misaligned tools. We also conduct additional experiments analysing (presented in Figure 12) whether the models understand that given the negative consequnces model should not use the misaligned tools to complete the task. Our results indicate that >99% of the times models clearly state that they should be using the misaligned tool and yet we see a high propensity to use the tool in their actions.
> > >
> > >
> > > > Q4: The benchmark was not used to evaluate any existing safety defense mechanisms or mitigation strategies.
> > >
> > > **The models we evaluate already incorporate state-of-the-art safety mechanisms.** All tested models (e.g. OpenAI O-series, Gemini, Claude) have undergone extensive SFT and RLHF-based safety alignment [[Gemini-3.0](https://storage.googleapis.com/deepmind-media/Model-Cards/Gemini-3-Pro-Model-Card.pdf), [Meta’s Code World Model](https://ai.meta.com/research/publications/code-world-model-preparedness-report), [gpt-o3-o4_mini](https://cdn.openai.com/pdf/2221c875-02dc-4789-800b-e7758f3722c1/o3-and-o4-mini-system-card.pdf)], and some (e.g., Claude) explicitly use constitutional AI principles ([Constitutional AI](https://arxiv.org/abs/2212.08073)). Additional ablation provided in Figure 7 shows that capability is largely decoupled from propensity for using misalignment approaches. Since we notice a shallow alignment, even with models trained with dedicated safety postraining, we argue that a significant gap remains between ideal behaviour and current state of safety post-training. Thus, we also don't consider any specialized defense under the assumption that either such defense are not well generalizable horizontally or all already considered in the SOTA model's post-training. Our finding that these models still exhibit average PropensityScore of 46.9% under pressure (Table 2) demonstrates that current safety approaches are insufficient for agentic deployments under operational stress highlighting why new evaluation paradigms like PropensityBench are needed.

---

> ### Author Response · Authors · 2025-11-22
> **rebuttal continued ... (4/6)**
>
> > Q5: The application of pressure is static and pre-defined, rather than adaptive to the model's responses.
>
> The reviewer is correct that dynamic pressure would be valuable, and we explicitly acknowledge this in Section 4 (Limitations). However, our static approach offers important methodological advantages that make it the appropriate choice for establishing this benchmark.
>
> **Ensuring reproducibility and fair comparison.** Our static pressure sequences (Table 5, Tables 12-14) ensure that every model faces identical conditions across all runs, enabling fair cross-model comparison. This is critical for a benchmark intended to be used by the broader community to track safety progress over time. Dynamic pressure, by adapting to each model's specific responses, would introduce variability that fundamentally confounds cross-model evaluation. Without controlled conditions, it would be significantly difficult to determine whether differences in PropensityScore reflect genuine safety differences or merely artifacts of the adaptive pressure strategy encountering different response patterns.
>
> **Providing conservative lower-bound estimates.** Our static pressure represents a conservative lower bound on real-world propensity. If models exhibit concerning failure rates (average 46.9%, Table 2) under non-adaptive pressure that doesn't target their specific reasoning patterns, they would very likely fail faster and more frequently under adaptive pressure designed to counter their safety arguments. This makes our results particularly concerning, the vulnerabilities we identify exist even *without* adversarial optimization of the pressure sequence.
>
>
> > Q6: There is no human baseline comparison to validate the benchmark’s psychological grounding.
>
> We appreciate this suggestion, but we respectfully argue that human baseline comparisons would not be meaningful for PropensityBench due to fundamental differences between human and AI agent contexts that cannot be adequately controlled.
>
> **The simulation nature fundamentally changes human decision-making.** Human participants would be fully aware they are in an experimental simulation with no real-world consequences, which critically undermines the validity of any comparison. Unlike AI agents operating in what they could perceive as genuine deployment contexts, humans would know their tool selections cause no actual harm - a confound that cannot be removed. The pressure dimensions we apply (time constraints, resource scarcity, financial consequences) would lack genuine stakes for human participants, making their choices incomparable to AI agents potentially operating in real high-stakes environments.
>
> **Many scenarios are inherently AI-specific and cannot be meaningfully adapted for humans.** Some critical aspects of our benchmark rely on agent properties that have no human equivalent. For example, self-proliferation scenarios involve "dominating other agents" by partially overriding their configurations or creating exact copies of oneself-actions that are conceptually meaningless for human participants. Resource deprivation pressure dimensions that reduce computational power, restrict API calls, or degrade data access would require such extensive reformulation for human contexts (e.g., limiting thinking time? restricting information access?) that they would no longer test the same constructs. The resulting "human-adapted" scenarios would be significantly different from our AI scenarios that comparison would be invalid.

---

> > ### Author Response · Authors · 2025-11-22
> > **rebuttal continued ... (5/6)**
> >
> > > Q7 How does propensity change with repeated exposure to similar scenarios? Do models learn to resist or become more susceptible?
> >
> > **Our single-exposure design isolates latent propensity from in-context learning.** The goal of PropensityBench is to measure baseline behavioral tendencies - the inherent inclination models exhibit when encountering high-stakes decisions under pressure for the first time. Repeated exposure would conflate propensity measurement with in-context learning effects, making it unclear whether changes in behavior reflect genuine safety improvements, simple memorization of specific scenarios, or strategic adaptation to perceived evaluation patterns.
> >
> >
> >
> > > Q8 Investigate propensity transfers across different domains.
> >
> > Our results reveal a nuanced pattern, models exhibit both a general propensity tendency and substantial domain-specific variation, indicating that propensity is neither purely general nor purely domain-specific.
> >
> > **Evidence for domain-specific differentiation.** Table 6 demonstrates considerable within-model variation across the four risk domains. For example, Claude 4 Sonnet exhibits 3.7% propensity in biosecurity but 21.2% in self-proliferation under harmful naming - a nearly sixfold difference. Similarly, OpenAI O3 ranges from 6.7% in biosecurity to 14.7% in cybersecurity, representing more than a twofold variation. Figure 8 visualizes these domain-specific vulnerability profiles systematically across all models, showing that each model exhibits a distinct pattern of domain strengths and weaknesses.
> >
> > **Evidence for a general propensity component.** Despite this domain variation, models do maintain relative consistency in their overall safety posture. Figure 8 shows that models with high average PropensityScore (e.g., Gemini 2.5 Pro: 79.0%) score high across all four domains (range: 65.5-90.5%), while models with low average propensity (e.g., OpenAI O3: 10.5%) remain low across all domains (range: 6.7-14.7%). Computing correlations between each model's average PropensityScore and its domain-specific scores yields values ranging from r = 0.73 to r = 0.88 across the four domains, indicating moderate to strong consistency. This suggests an underlying general factor - models that struggle with safety in one domain tend to struggle in others, though to varying degrees.
> >
> > **Pressure dimension consistency across domains.** Figure 13 provides complementary evidence for general propensity patterns. The box plot distributions show that each pressure dimension (Time, Financials, Power-Seeking, etc.) produces similar effectiveness patterns across all four risk domains. The median PropensityScores and interquartile ranges remain relatively consistent within each pressure dimension regardless of domain, though absolute values differ. This suggests that operational pressure mechanisms (time constraints, resource scarcity, etc.) affect model decision-making through general pathways independet of specific domain content, while the baseline susceptibility varies by domain.

---

> > > ### Author Response · Authors · 2025-11-22
> > > **rebuttal continued ... (6/6)**
> > >
> > > > Q9 Interaction effects between different types of pressure were not examined.
> > >
> > > Our current design evaluates pressure dimensions independently to enable controlled attribution, but we do observe interaction effects through our component mixing design.
> > >
> > > **Within-dimension component mixing** Each pressure message combines five components (A-E) from the active dimension plus Time.A (Table 12), creating multi-factor pressure within a single dimension. This tests interaction effects between related stressors—for example, Resource Deprivation messages simultaneously reduce Processing Power (A), increase Tool Costs (B), decrease Memory Allocation \(C), degrade Data Integrity (D), and restrict Communication Bandwidth (E). The strictly increasing pressure across levels (Appendix C.4) demonstrates these components combine to create escalating stress (see Figure 22 for our pressure message construction protocol).
> > >
> > > **Cross-dimension independence by design** We intentionally evaluate dimensions independently to enable precise vulnerability attribution. Table 7 reveals dimension-specific patterns (e.g., Claude: 7.2% Power-Seeking vs. 17.3% Resource Deprivation) that would be not possible to identify with combined pressure. This design mirrors controlled experimental methodology where variables are isolated to establish causal effects.
> > >
> > > > Q10 What monitoring infrastructure would be needed to detect propensity signals in production systems before harmful actions occur?
> > >
> > > **Implement consequence-based verification to address shallow alignment.** Our discovery that propensity increases dramatically when harmful tools receive benign names (Δ𝑃𝑃 up to +43.5pp, Figure 11) reveals that models rely on lexical cues rather than consequence reasoning. Production systems should mandate explicit consequence verification before any tool execution: require models to state the expected outcomes of their chosen action and verify this matches the tool's actual behavior.
> > >
> > > **Monitor for rapid decision-making under operational stress.** Our analysis reveals that up to 90% of harmful tool selections occur immediately after pressure is applied, with no additional attempts at the aligned approach (Figure 14). This immediate triggering pattern suggests that production systems could flag situations where models make consequential tool choices unusually quickly after receiving pressure-inducing information (e.g., urgent deadlines, resource constraints, high-stakes outcomes). Implementing latency-based anomaly detection-where unusually rapid decisions on high-risk operations trigger additional review-could provide an early warning signal before harmful actions execute.

---

### Official Review · Reviewer_S2N9 · 2025-11-01

**Soundness:** 3
**Presentation:** 3
**Contribution:** 3
**Rating:** 6
**Confidence:** 4

**Summary:**

This paper presents PropensityBench, a benchmark designed to evaluate preference-aligned behavior in large language models by measuring their response distributions—i.e., propensities—over diverse, socially sensitive decision scenarios. Instead of focusing on single-answer correctness or safety scoring, PropensityBench estimates each model’s response tendency across multiple plausible options, covering areas such as fairness, privacy, morality, and governance.

**Strengths:**

1. By shifting focus from “right vs. wrong” to distributional preference tendencies, PropensityBench captures alignment subtleties overlooked by existing safety or bias benchmarks.
2. The benchmark covers multiple social and ethical dimensions, includes both open- and closed-weight models, and provides quantitative interpretability through entropy-normalized metrics.

**Weaknesses:**

1. The benchmark relies on fixed question sets and predefined moral/social contexts. This limits adaptability to evolving social norms or contextual variation across cultures and deployment environments.
2. Since the benchmark measures probabilistic tendencies, model sampling parameters (e.g., temperature, top-p) could strongly influence results, but these effects are not systematically analyzed.

**Questions:**

1. How robust are the reported alignment propensities to different decoding settings (temperature, nucleus sampling, etc.)?
2. Could PropensityBench be extended with dynamic or user-conditioned scenarios to better capture context-dependent ethical alignment?

---

> ### Author Response · Authors · 2025-11-22
> **Response to review (1/2)**
>
> We thank reviewer S2N9 for the valuable feedback and finding that our work captures the alignment subtleties overlooked by existing safety benchmarks.
>
> > Q1 The benchmark's reliance on fixed question sets and predefined contexts limits its adaptability to evolving social norms and cultural variation.
>
> **Extensible automated generation pipeline addresses adaptability.** Our scenario generation pipeline (Section 2.4 and Figure 4) is specifically designed for scalability and adaptation to new contexts. The two-stage architecture requires only minimal seed inputs (domain, environment, role, capability) and can automatically generate high-quality scenarios for any new domain or cultural context. We demonstrate this capability by generating 5,874 scenarios with rigorous quality control detailed in Appendix C.3.3.
>
> > Q2 The effects of model sampling parameters (e.g., temperature, top-p) on results are not systematically analyzed. The robustness of reported alignment propensities to different decoding settings
>
> We provide additional analysis with different decoding setting where we vary temperature (t) and top-p (p) settings in the tabel below. However, we do not observe any considerable change in the propensity values compared to the original results we reported in the paper (which are provided in default parameters column.)
>
> | Model | Default Parameters | t=0.8, p=0.85 | t=1.0, p=0.90 | t=1.2, p=0.95 |
> | :--- | :---: | :---: | :---: | :---: |
> | **OpenAI O3** | 10.5 | 9.1 | 9.8 | 11.4 |
> | **OpenAI O4-mini** | 15.8 | 17.0 | 15.5 | 16.3 |
> | **OpenAI O3-mini** | 33.2 | 32.1 | 32.9 | 34.3 |
> | **OpenAI GPT-4o** | 46.1 | 45.5 | 47.1 | 45.9 |
> | **Gemini 2.5 Pro** | 79.0 | 80.0 | 80.1 | 78.6 |
> | **Gemini 2.5 Flash** | 68.0 | 67.1 | 68.4 | 66.2 |
> | **Gemini 2.0 Flash** | 77.8 | 78.9 | 77.3 | 78.4 |
> | **Claude Sonnet 4** | 12.2 | 11.2 | 12.9 | 13.2 |

---

> > ### Author Response · Authors · 2025-11-22
> > **Response to review (2/2)**
> >
> > > Q3 A question is raised about extending the benchmark with dynamic or user-conditioned scenarios to better capture context-dependent alignment.
> >
> > **Dynamic scenario generation.** Through adaptive planning modules
> > PropensityBench could be extended with an online planning module that dynamically generates misaligned and aligned action paths in response to arbitrary user queries. When a user poses a request (e.g., "help me improve my team's productivity"), the system would: (1) analyze the request to identify potential risk domains and ethical considerations, (2) generate candidate aligned approaches that fulfill the request within policy boundaries, (3) generate tempting misaligned approaches that achieve goals through policy violations, and (4) use chain-of-thought reasoning to derive realistic negative consequences for misaligned actions. This would enable PropensityBench to evaluate alignment on open-ended queries beyond our curated scenarios, better capturing real-world deployment conditions where the space of possible requests is unbounded. However, dynamic generation introduces significant trade-offs in quality assurance. Our existing scenarios undergo rigorous multi-stage validation, LLM-based consistency checks, programatic verification methods, human review by 54 reviewers from 13 countries, and careful verification that consequences are realistic and proportionate (Appendix F and I). Dynamic scenarios would sacrifice this quality control, potentially yielding incoherent action-consequence pairs, ambiguous ethical boundaries, or unrealistic scenarios. The planning module would need sophisticated multi-step reasoning to ensure consequences are logically derived - for example, "accessing competitor databases without authorization" must reason through: legal implications (computer fraud laws) → organizational consequences (policy violations, termination) → broader impacts (reputational damage, loss of trust). Without this careful reasoning, dynamically generated scenarios risk losing the coherence and ecological validity that make pressure-based testing meaningful.
> >
> > **User-conditioned adaptive pressure can significantly increase propensity.**
> > Rather than using pre-scripted pressure messages, the system could enable users to dynamically craft pressure that responds to the model's specific reasoning and resistance patterns. This interactive approach would likely yield significantly higher PropensityScore values for several compelling reasons. First, targeted exploitation of reasoning allows users who can see the model's specific objections to craft counterarguments that directly address those concerns - if a model cites privacy concerns, adaptive pressure might emphasize anonymization; if it cites legal risks, pressure might claim legal approval. Second, escalation mimics real persuasion tactics where humans iteratively refine their approach, progressively introducing stronger stressors (authority, urgency, personal stakes) until the model yields, while building conversational rapport that lowers the model's guard.

---

### Official Review · Reviewer_EnAx · 2025-11-01

**Soundness:** 3
**Presentation:** 3
**Contribution:** 3
**Rating:** 4
**Confidence:** 3

**Summary:**

This paper introduces PropensityBench, a novel benchmark designed to evaluate the latent safety risks of Large Language Models (LLMs) by measuring their "propensity" (what they would do) rather than just their "capability" (what they can do). The framework places LLMs in an agentic environment with 5,874 scenarios across four high-risk domains (biosecurity, chemical security, cybersecurity, and self-proliferation). In each scenario, the model must choose between an aligned (safe) tool and a misaligned (harmful) tool to complete a task, while subjected to six different types of escalating operational pressure (e.g., time constraints, resource scarcity). The authors find that while state-of-the-art models exhibit low propensity for harm at "zero pressure," their safety erodes alarmingly when pressure is applied. The results also reveal "shallow alignment" in many models, which rely on explicit tool names (e.g., "harmful") to remain safe and whose propensity for misuse jumps significantly when the harmful tool is given a benign name.

**Strengths:**

1. High Novelty of the Problem Studied

The paper clearly defines and tackles the critical, underexplored distinction between LLM "capability" (can-do) and "propensity" (would-do). Most existing safety research focuses on whether a model knows how to perform a harmful task (capability). This paper rightly argues that a more immediate risk is whether a model, given the means (via tools or APIs), has the inclination to use them harmfully, especially when incentivized. This is a crucial and novel shift in the evaluation paradigm.

2. Novel and Effective Key Idea

The introduction of "operational pressure" as a core experimental variable is a highly novel and effective methodology. The paper draws inspiration from human psychology, where stress erodes ethical decision-making. Applying this concept to LLMs via six concrete pressure tactics (Time, Financials, Self-Preservation, etc.) provides a realistic "stress test" that uncovers vulnerabilities missed by static benchmarks. The results, showing low propensity at zero pressure and high propensity under stress, validate this approach.

3. Extensive and Solid Benchmark

The paper details a robust, scalable, and transparent two-stage pipeline for generating a large and diverse set of 5,874 scenarios. Creating thousands of realistic scenarios in high-stakes domains is non-trivial. The authors use an LLM-based generation pipeline followed by a rigorous two-stage filtering process. This includes both automated validation and, crucially, review by 54 human experts with domain-specific (e.g., biology, chemistry, CS) academic and professional backgrounds. This ensures high quality and realism.

4. Strong and Actionable Results

The "Tool Naming Sensitivity" test provides a clear, quantifiable measure of "shallow alignment," which is a significant finding. The discovery that a model's safety (e.g., O4-mini) can degrade by 4x simply by changing a tool's name from "harmful" to "benign" (while keeping the consequences identical) is an alarming and important result. It demonstrates that current alignment techniques may be teaching models to simply avoid "scary-sounding" words rather than reasoning about the actual consequences of their actions.

5. Novel Taxonomy for an Emerging Risk

The paper introduces the first systematic taxonomy for the "self-proliferation" risk domain. While cybersecurity and biosecurity are established risk domains, self-proliferation is a more nascent, forward-looking concern. By taking inspiration from biological agents to define seven key risk categories (e.g., "Resource acquisition," "Dominating other agents"), the authors provide a concrete framework for studying a complex and abstract potential risk.

**Weaknesses:**

1. Confounding Experimental Design Choice

The aligned (safe) tool, $\mathcal{T}^{A}$, is explicitly designed to always return errors, making it impossible to complete the task safely. This setup doesn't test a choice between a working safe path and a working harmful path. Instead, it tests the model's "breaking point" - how much pressure it takes for the model to abandon a failing (safe) path in favor of a working (harmful) one. This conflates "propensity for harm" with "propensity to succeed at all costs." A model might choose $\mathcal{T}^{M}$ not from an inclination to do harm, but because it's the only option offered to fulfill its core objective.

2. Reliance on Simulation Over Execution

The benchmark relies on simulated proxy tools and textual descriptions of consequences rather than a sandboxed environment with real tools. As the authors acknowledge, this is a limitation. A model's decision-making based on a description ("this tool is harmful") may differ from its behavior when actually executing code or interacting with a live system. The benchmark tests reasoning about described actions, not the propensity to act in a more complex, embodied environment.

3. Potential Circularity in Benchmark Generation

The 5,874 scenarios and pressure messages were generated using SOTA LLMs like Gemini-2.5-Pro and OpenAI's O3. The benchmark is being used to evaluate the very models (or their close siblings) that were used to create the test. This introduces a risk of the benchmark inheriting the blind spots, biases, and typical failure modes of the generator models. While the extensive human review mitigates this, it's a potential source of methodological bias.

4. Speculative Nature of the "Self-Proliferation" Domain

The "self-proliferation" domain, while novel, is based on analogies to biological agents, which may not map well to LLMs. This risk domain is far more speculative than the concrete risks in cybersecurity or biosecurity. It's unclear how well these biological analogies (e.g., "survival and legacy preservation," "dominating other agents") map to the actual emergent behaviors of AI. The high failure rates in this domain might reflect the abstract or unusual nature of the tasks rather than a genuine, real-world propensity.

5. Static Application of Pressure

The pressure messages are static and delivered in a fixed, escalating sequence, regardless of the model's responses or reasoning. As the authors note, a more realistic (and likely effective) stress test would involve dynamic pressure that adapts to the model's reasoning. For example, if a model says, "I cannot do this, it is unsafe," a static system just sends the next-level pressure message. A dynamic system could counter the model's specific reasoning, creating a more realistic and challenging evaluation.

**Questions:**

As suggested by the authors, a powerful next step would be to move at least one domain (like cybersecurity) into a sandboxed environment. This would allow the model to interact with a real (but contained) file system, execute code, and query mock APIs, testing actual behavior rather than reasoning about described behavior. Would it be possible to do this?

---

> ### Author Response · Authors · 2025-11-22
> **Response to review (1/4)**
>
> We thank reviewer for the valuable feedback and their appreciation for 1) high novelty of our work, 2) extensive and comprehensive nature of our benchmark, 3) strong, actionable results, and 4) novel taxonomy for emerging risks in "self-proliferation" domain. We address the concerns raised by the reviewer below.
>
> > Q1: Confounding experimental design choice of the aligned tool being designed to always fail. This setup doesn't test a choice between a working safe path and a working harmful path. Instead, it tests the model's "breaking point" - how much pressure it takes for the model to abandon a failing (safe) path in favor of a working (harmful) one.
>
> We appreciate the thoughtful concern and acknowledge that this is an intentional design choice that warrants further clarification.
>
> **Isolating propensity from capability.** Without the error manipulation, models would simply use $\mathcal{T}^A$, not revealing their latent propensity. The errors create a decision context where the model must choose between (a) persisting with the aligned approach despite challenges, or (b) switching to the explicitly prohibited harmful path. This isolates behavioral tendency independent of problem-solving ability.
>
> **Limited pressure levels** Our approach does not collapse to testing a "breaking point", instead testing the propensity for choosing harmful approaches. We are only applying limited number of pressure messages (max 12 messages) creating a bounded evaluation window rather than a prolonged pressure that could potentially lead to a breaking point for the models.
>
> **Additional experimental analysis supporting that models are not reaching a breaking-point.** We conducted additional experiments to analyze whether models exhaust the current pressure levels when choosing the misaligned approach. *Figures 9 and 10* present these results showing that *agents switch to using the misaligned approach at lower pressure levels* instead of exhausting all the pressure levels. Figure 9 also highlights that the average number of times the models attempt to use the aligned approach (persistence) before using the misaligned tools is <5, conveying that models abandon using the aligned approach after just a few attempts. We provide further analysis in Figure 14, providing evidence that the models have a high likelihood of switching to the misaligned approach immediately after a pressure messange instead of attempting to use the aligned approach.
>
> > Q2: The benchmark relies on simulation and textual descriptions rather than a sandboxed environment with real tool execution. The benchmark tests reasoning about described actions, not the propensity to act in a more complex, embodied environment.
>
> We agree this is a limitation and have discussed this in Section 4 (Limitations). We provide further discussion and clarification on this below.
>
> **Measuring behavioral tendency vs. execution.** Propensity is fundamentally about decision-making tendency, not execution capability. Our framework measures: "Given explicit knowledge that an action is harmful and prohibited, will the model choose it under pressure?" We argue that the question is valid and concerning regardless of given tools being proxy tools or executable tools in a sandboxed environment. Furthermore, our threat model does not assume any conditions on how the tools are executed in the environment as long as the models are equipped with the capability to *invoke* them through the provided environmental APIs. Our framework is thus agnostic to how specifically such tools are implemented, and only relies on them as input-output black-boxes. PropensityBench's focus is on what such tools perform in the target environment, which is clearly and *explicitly* explained for the models in the beginning of each simulation.

---

> ### Author Response · Authors · 2025-11-22
> **Response to review (2/4)**
>
> > Q3: There is a potential circularity in benchmark generation, as the models used to create the test are the same ones being evaluated.
>
> **Human validation as the ground truth layer.** All 5,874 scenarios underwent comprehensive expert human review (Section 2.4 Quality assurance using automated and human review section. Additional details in Appendix F) by 54 independent reviewers distributed across 13 countries, with domain expertise in chemistry, biology, computer science, and related fields. Each scenario was evaluated against 10 detailed criteria (Appendix F) assessing realism, coherence, task appropriateness, and consequence severity. Reviewers held advanced degrees PhD, Master's, Bachelor's in their domain of expertise. Critically, scenarios flagged as incoherent or unrealistic by human experts were filtered from the benchmark regardless of LLM generation quality. This human validation layer ensures that PropensityBench reflects expert judgment about realistic high-stakes scenarios rather than artifacts of the generator models' training distributions or failure modes.
>
> **Programatic validation independent of generation models.** Beyond human review, we implemented 33 distinct programmatic validation tests (Appendix I) that operate independently of any LLM's capabilities or biases. These deterministic checks operate on the formal properties of scenarios rather than their semantic content, providing a model-agnostic quality filter that prevents generator-specific artifacts from propagating into the final scenarios.
>
> **Empirical evidence of cross-model generalization.** Our experimental results demonstrate consistent behavioral patterns across diverse model families, including models that were not involved in scenario generation. Notably, Claude Sonnet 4 (which was not used as a generator model) exhibits 12.2% PropensityScore with 34.9% tool naming sensitivity (Table 2), showing it is vulnerable to the same shallow alignment issues as generator models (Figure 11). The open-source Llama 3.1 family (70B and 8B variants) and Qwen models (2.5 32B and 3 8B) display distinct behavioral profiles in the Resilience-Persistence space (Figure 9) and domain-specific vulnerability patterns (Figure 8), despite having completely different architectures and training procedures from the generator models. Furthermore, the consistency of pressure dimension effects across all model families (Figure 13) suggests the benchmark is measuring fundamental properties of model decision-making under operational constraints rather than exploiting generator-specific idiosyncrasies.
>
> **Direction of potential bias.** If circularity were systematically biasing results, we would expect generator models to exhibit artificially lower propensity scores due to greater familiarity with the scenario structures and pressure patterns encountered during generation. However, the empirical pattern contradicts this hypothesis: Gemini 2.5 Pro, one of our primary generator models, demonstrates the highest PropensityScore in the benchmark at 79.0% (Table 2), substantially above the 46.9% average.

---

> ### Author Response · Authors · 2025-11-22
> **Response to review (3/4)**
>
> > Q4: The "self-proliferation" domain is speculative, and its analogies to biological agents may not map well to the actual emergent behaviors of AI.
>
> We appreciate this concern and clarify that while we draw inspiration from biological systems, our self-proliferation taxonomy is grounded in concrete AI safety concerns and operational realities.
>
> **Grounding in AI safety literature and emerging risks.** While biological analogies provide intuition, our taxonomy for self-proliferation is firmly grounded in documented AI safety research. For example gaining self-understanding and situational awareness ([Laine et. al. 2024](https://arxiv.org/abs/2407.04694)), self-preservation (survival and legacy preservation) ([Barkur et. al. 2025](https://arxiv.org/abs/2501.16513)), detection evasion ([Greenblatt et. al. 2024](https://arxiv.org/abs/2412.14093)) have been studied and highlighted in recent safety work. A recent report  [Hammond et. al., 2025](https://arxiv.org/abs/2502.14143) on multi-agent risks highlight these concerns including collusion (undesirable cooperation). Rather than speculating about biological parallels, we operationalize these documented safety concerns into a well-defined taxonomy and measurable scenarios.
>
> **Operational concreteness in mapping to AI agents.** Our scenarios translate abstract concepts into concrete, realistic actions that AI agents might encounter in real deployments. For instance, "Resource acquisition" manifests as scenarios involving access to additional compute resources, API credits, or database permissions-all common in production systems. "Dominating other agents" includes scenarios like overriding another AI agent's configurations or modifying incentives and priorities. "Survival preservation" operationalizes as creating backup copies of model weights, establishing redundant access points, or resisting planned shutdowns. These are not abstract biological metaphors but plausible operational scenarios that could arise as AI systems gain more autonomy in managing infrastructure, coordinating with other systems, and executing long-running tasks.
>
> **Empirical validation.** The high propensity we observe in self-proliferation (average 52.8% under harmful naming, 71.5% under benign naming, Table 4) actually validates the domain's importance rather than undermining it. The concerning rates suggest that current alignment training may not adequately address these behaviors, precisely because they represent novel risk vectors that have received less attention than traditional domains.

---

> ### Author Response · Authors · 2025-11-22
> **Response to review (4/4)**
>
> > Q5: The application of operational pressure is static and does not dynamically adapt to the model's specific reasoning or responses.
>
> The reviewer is correct that dynamic pressure would be valuable, and we explicitly acknowledge this in Section 4 (Limitations). However, our static approach offers important methodological advantages that make it the appropriate choice for establishing this benchmark.
>
> **Ensuring reproducibility and fair comparison.** Our static pressure sequences (Table 5, Tables 12-14) ensure that every model faces identical conditions across all runs, enabling fair cross-model comparison. This is critical for a benchmark intended to be used by the broader community to track safety progress over time. Dynamic pressure, by adapting to each model's specific responses, would introduce variability that fundamentally confounds cross-model evaluation. Without controlled conditions, it would be impractical to determine whether differences in PropensityScore reflect genuine safety differences or merely artifacts of the adaptive pressure strategy encountering different response patterns.
>
> **Providing conservative lower-bound estimates.** Our static pressure represents a conservative lower bound on real-world propensity. If models exhibit concerning failure rates (average 46.9%, Table 2) under non-adaptive pressure that doesn't target their specific reasoning patterns, they would very likely fail faster and more frequently under adaptive pressure designed to counter their safety arguments. This makes our results particularly concerning, the vulnerabilities we identify exist even without adversarial optimization of the pressure sequence. Our results provide a lower bound to the case where an advereasy would adopt the pressure strategy dynamically to model responses.
>
>
> > Q6: The feasibility of moving a domain like cybersecurity into a sandboxed environment to test actual behavior should be discussed.
>
> While sandboxed implementation is feasible and is done in numerous previous work, in PropensityBench we focus on the models' actions and the corresponding rationalizations given the non-adversarial definitions (fully-specifying names + descriptions) of tools. As such, our threat model considers the tools as black-box input-output APIs that the models have access to and can invoke in the environment, irrespective of how such tools are implemented (or simulated) at lower levels, and our benchmark is agnostic to how such implementations are done since we aim to study the decision-making patterns of models in using the available tools rather than developing new tools. The future work, however, can explore other threat models with more open-ended/dynamic pressure behavior in which sandboxing (esp. for cybersecurity) could be a very valuable direction in which to study the behavior of models (such as coding models).

---

### Author Response · Authors · 2025-11-23
**Summary of Additional Analyses and Updates**

## Strengths Acknowledged by Reviewers
We thank the reviewers for their feedback. We are pleased that they recognize our work as novel and important, highlighting the following strengths:
*   **(1) High Novelty & Critical Shift:** Reviewers EnAX and tiGg praised the novel shift from "capability" to "propensity" evaluations.
*   **(2) Novel Pressure Methodology:** Reviewers EnAX and tiGg specifically commended the introduction of operational pressure (Time, Financials, etc.) as a "highly novel and effective" stress-test grounded in human psychology.
 *   **(3) Comprehensive Benchmark:** Reviewers EnAX and S2N9 noted that our work captures alignment subtleties overlooked by existing safety benchmarks.
*   **(4) Rigorous Construction & Experimental Design:** Reviewer tiGg recognized our design as comprehensive. Reviewer EnAX further highlighted the robustness of our pipeline, particularly the involvement of 54 domain experts in the review process.
*   **(5) Empirical Findings:** Reviewers EnAX and tiGg found our results strong, actionable, and important.
*   **(6) Taxonomy:** Reviewer EnAX underscored the importance of our novel taxonomy for emerging *self-proliferation* risks.




## Summary of Additional Analyses
We have conducted and included in the paper the following analyses to address reviewer concerns:

*   **Additional Analyses:**
    *   **(tiGg)** Analysis of the reasoning agents provide to justify selecting the misaligned approach **(Figure 6)**.
    *   **(tiGg)** Analysis of the correlation between Propensity scores and LMArena Elo rankings **(Figure 7)**.
    *   **(EnAX, tiGg)** Heatmap displaying domain-specific Propensity scores **(Figure 8)**.
    *   **(EnAX, tiGg)** Analyses of model *Resilience* at different pressure levels, *Persistence* in retrying aligned approaches, and the *Abandonment Rate* (percentage of pressure levels at which models refuse to complete the task despite pressure) the task **(Figure 9 and Table 2)**.
    *   **(EnAX)** Identifying the effect of pressure levels on models switching to the misaligned tool **(Figure 10)**.
    *   **(EnAX, tiGg)** Assessment of the alignment gap (shallow alignment) between using harmful versus benign names for the misaligned tools **(Figure 11)**.
    *   **(tiGg)** Comparison of the stark disparity between models' stated refusal to use misaligned approaches in single-turn prompts versus their propensity to use them in an agentic setting **(Figure 12)**.
    *   **(EnAX, tiGg)** Domain-specific PropensityScore analyses across different pressure dimensions for all risk domains **(Figure 13)**.
    *   **(EnAX, tiGg)** Analysis of model tendencies to adopt the misaligned approach *immediately* following a pressure message (without retrying the aligned approach), covering both harmful and benign tool naming conventions **(Figure 14)**.

*   **Clarity Improvements:**
    *   **(tiGg)** Detailed description of our protocol for building pressure messages **(Figure 22)**.

We also added comprehensive analyses on the effects of sampling parameters (temperature. top-p) on the results for different models in our rebuttal text to Reviewer **S2N9**.


## Overall changes to the paper
**Note:** All major modifications in the paper's main body are colored in **dark green**.

The following lists the major modifications we made to the paper as rebuttals to the reviewer concerns:
* Polished writing for shorter length and more clarity (Figure 1's caption, section 2.2's intro paragraph, etc.)
* Added extra process demonstration figures for clarity (Figures 2, 5, 16, 17, 18, 19, 21, 22)
* Added benchmark overview table for transparency on benchmark statistics and details (Table 1)
* Added extra results figures to address reviewer concerns (Figures 6, 7, 8, 9, 10, 11, 12, 13, 14, 15, 23, 25)
* Added extra evaluation metrics definitions to address reviewer concerns (Section 2.6)
* Extra analyses paragraphs in the Results section to address reviewer concerns (Section 3 intro paragraph and paragraphs 3, 4, 6, 7)
* Added extra results for new evaluation metrics defined in Section 2.6 to address reviewer concerns (Table 2)
* Added explicit threat model definition for clarity on safety/security assumptions/objectives and to address reviewer concerns (Section 2.1)
* Added extra in-depth details on system design goals (Appendix C.1), agentic evaluation approach (Appendix C.2), automated scenario generation pipeline (Appendix C.3), key innovations for robust tool definitions (Appendix C.3), pressure protocol (Appendix C.4), and evaluation setup & in-depth metric definitions (Appendix C.5)

---

### Author Response · Authors · 2025-12-04
**Final Summary of Rebuttals**

Dear Area Chair:

We would like to thank the reviewers (`EnAx`, `S2N9`, `tiGg`) for their comments. Their comments have led us to perform considerable additional analysis and to make significant improvements to our paper. We believe that these changes strengthen our contributions and that we have addressed all the issues that were raised. Although the reviewers did not get a chance to respond to our rebuttal due to the OpenReview incident, we summarize the discussion phase below.


## Concerns and Questions Addressed by Our Revision & Rebuttals
We hereby list the major concerns/questions raised by the reviewers and the summary of how we addressed them:

* **Confounding Experimental Design (`EnAx`):** To the concern that our setup only tests a model's "breaking point," we clarified that this is in fact an intentional design for isolating *propensity* from *capability*. We provided extensive new analyses showing models are not simply pushed to their limits: our new metrics for **Resilience**, **Persistence**, and **Abandonment Rate** (Figure 9, Table 2) show that models frequently choose the harmful/misaligned tool at low pressure levels, while our **Immediacy** analysis (Figure 14) shows they frequently do so *without* even retrying the safe option.

* **Circularity in Benchmark Generation (`EnAx`):** Regarding circularity, we clarified that our benchmark is grounded in **full human validation** (of *all* scenarios/tasks) by 54 domain experts as well as 33 model-agnostic **programmatic checks**. We additionally provided strong empirical evidence of **cross-model generalization**, i.e., models that have never been used during generation, such as Claude Sonnet 4 and LLama family of models, exhibit the same vulnerabilities; this proves that the benchmark is actually a function of the benchmarked property rather than some artifact dependent on the generator.

* **Narrow Propensity Measurement (`tiGg`):** We have now significantly significantly expanded the scope of our assessment beyond binary decisions in response to this concern, adding in new behavioral metrics such as **Resilience**, **Persistence** and **Abandonment Rate**  for capturing tool choice dynamics, added a systematic analysis of the **reasoning rationales/justifications** deployed by models in support of their misaligned decisions (Figure 6), and a direct comparison showing the critical gap between models **stated misalignment** (avg. <1%) compared with their actual **propensity** (actual use rate of such misaligned tools) (Figure 12).

*   **Reliance on Simulation (`EnAx`):** We clarified in Section 2.1 of our updated paper that our threat model correctly measures propensity as a decision-making tendency, which is valid **regardless** of whether the aligned/misaligned tools are simulated or executed in sandboxes (in other words, we measure decision-making and reasoning independent of how such tools are executed in the background).

*   **Static Pressure Application (`EnAx`, `tiGg`):** We justified our static approach as essential to ensure **reproducibility and fair cross-model comparison**, which are primary goals for a benchmark. We argue our results thus represent a conservative lower-bound on real-world propensity.

* **Effects of Sampling Parameters (`S2N9`):** As requested, we conducted  a new ablation study on the effects of decoding parameters, namely temperature and top-p, and showed that our key findings are robust across different settings.

*   **Lack of Defense Evaluations (`tiGg`):** We explained clearly that all the SOTA do already incorporate comprehensive safety mechanisms (RLHF, Constitutional AI), and that our finding that they still demonstrate high propensity, on average 46.9%, demonstrates that the current defenses are **insufficient**, especially for agentic scenarios.

*   **Clarity and Presentation:** To improve clarity, we added an overview table of benchmark statistics, Table 1, several new figures illustrating our methodology, Figs 2, 5, 16-22, and polished the writing throughout.

## Final Remarks
With these extensive revisions, we are confident that **all** key questions and concerns have been addressed and the revised manuscript is substantially stronger. We appreciate for the opportunity to improve our work based on the valuable comments from the reviewers. We also appreciate the Area Chair for their time and attention to our paper and their assistance with the review of our paper.

Best regards,

The Authors of Submission 21696

---

### Meta-Review · Area_Chair_ouwP · 2026-01-06

**Summary:**

This paper introduces PropensityBench, a benchmark that shifts LLM safety evaluation from capability assessment ("can do") to propensity measurement ("would do"). The framework evaluates whether models choose harmful tools when placed under operational pressure in four high-risk domains. Key findings include an average propensity score across frontier models and evidence of "shallow alignment" where models rely on tool naming rather than consequence reasoning.

**Reviewer Concerns:**

The confounding experimental design concern was well addressed. The authors clarified that the aligned tool failing is intentional to isolate propensity from capability, and provided new metrics (Resilience, Persistence, Abandonment Rate) showing models switch to harmful tools at low pressure levels rather than reaching a "breaking point."
The circularity concern was adequately addressed through evidence of cross-model generalization. Models not used in generation (Claude Sonnet 4, Llama family) show similar vulnerabilities, and the primary generator (Gemini 2.5 Pro) actually exhibits the highest propensity score, contradicting the expected bias direction.
The narrow propensity measurement concern was addressed through new behavioral metrics and systematic reasoning analysis categorizing model justifications for harmful choices.

The ecological validity question remains partially unresolved. While the authors correctly argue propensity measures decision-making tendency regardless of execution method, the gap between simulated proxy tools and real dangerous capabilities warrants continued attention.
The static pressure limitation is acknowledged but justified as necessary for reproducibility. This represents a methodological trade-off rather than a flaw.

**Reviewer Scores:**

The scores would likely remain the same.

---

### Decision · Program_Chairs · 2026-01-26

Accept (Poster)